# Does charitable giving reduce firms' willingness to invest in green innovation?

Hongpeng Wang[1,2,3]*

1 Business School, Hohai University, Nanjing, China, 2 Jiangsu Provincial Collaborative Innovation Center of World Water Valley and Water Ecological Civilization, Nanjing, China, 3 Jiangsu Decision Consulting Research Base (International Development of Enterprises), Nanjing, China

* 348557935@qq.com

## Abstract

While corporate charitable giving(CG) can help firms obtain external innovation resource support, it can also crowd out internal innovation resources. The purpose of this study is to clarify the mechanism of CG and government green subsidies(GS) on green innovation(GI). In this regard, we integrated signaling theory and principal-agent theory to provide a new theoretical perspective for simultaneously focus on the impact of external resource acquisition and internal resource allocation on GI. We conducted a threshold regression analysis on the balanced panel data of 863 listed companies of China from 2016 to 2019 to clarify the input boundary between the promoting and inhibiting effects of corporate CG on corporate GI. And we further explored the relationship between GS and GI under the effect of different CG thresholds. Our findings indicate that there is an inverted U-shaped threshold effect of CG on GI. The impact of GS on GI shows a decreasing marginal benefit as the intensity of CG increases. Based on the findings, we propose corresponding countermeasures for the management of enterprises and the government.

**Data Availability Statement:** The data underlying the results presented in the study are available from Water Economy Management Database (Hohai University, CN: http://dsp.hhu.edu.cn/data/).

**Funding:** This study was supported by the MOE (Ministry of Education in China) Project of

## Introduction

Green innovation (GI) is an innovation activity that concerns environmental protection and sustainable development in product development and design, a socially responsible behavior [1]. It is an essential way for enterprises to actively participate in China's "carbon neutral& emission peak" strategy. However, for companies, GI is both an opportunity and a challenge. On the one hand, green development is in line with the current era's development requirements, and the Chinese government attached great importance to subsidizing the enterprises which carry out GI. For example, the outline of China's 14th Five-Year Plan and the "2035 Vision" clearly indicates that enterprise innovation's primary position should be strengthened, and policy support should be provided for enterprises to carry out GI. Therefore, companies' proactive GI is expected to help them develop a competitive advantage in the industry early. However, on the other hand, with the recurrence of environmental uncertainties such as the COVID-19 epidemic, enterprises are facing increasingly serious survival problems. That makes many enterprises tired of dealing with short-term survival or development problems

Humanities and Social Sciences (Project No. 19YJC630155) to ZHANG Yang. The funder, Yang Zhang, played a role in concept construction, study design, decision to publish, and preparation of the manuscript.

**Competing interests:** The authors have declared that no competing interests exist.

and lack of attention to the investment of GI, and even produce "strategic innovation" behavior aimed at cheating the government green subsidies(GS) [2]. Therefore, it is essential to explore the corporate behavior that affects GI.

Enterprises need to maintain a steady input of resources to carry out GI activities, so they cannot do without the resource support from stakeholders. However, due to the asymmetry of information inside and outside the enterprise, external stakeholders usually choose whether to support the enterprise with resources or not based on a particular way of releasing signals from the enterprise [3]. Like a typical way for enterprises to actively assume social responsibility, corporate charitable giving(CG) not only helps enterprises send positive signals such as "responsible" and "financially sound" to consumers, investors, and other critical stakeholders through the media to help enterprises obtain market and external investment support [3, 4], but also shares the pressure of government work in epidemic fighting, disaster relief and poverty alleviation to gain government attention [5], making it easier for enterprises to obtain necessary resource support from the government such as in terms of subsidies and policies. Therefore, CG is considered by many scholars to be an essential influencing factor in the study of corporate innovation activities [3, 5]. This paper will explore the mechanism of CG on GI based on the characteristics of internal and external information asymmetry of firms.

Existing studies have not consistently concluded the relationship between CG and GI. Some scholars have argued that CG can release a positive image of the company to the outside world and help the company obtain more external resources to support [3], thus forming a resource-effect to promote GI. In contrast, scholars based on principal-agent theory(PAT) argue that corporate agents tend to take advantage of information asymmetry to seek private benefits, i.e., they make unreasonable allocation of corporate resources through charitable donations, diverting internal resources that should be used for GI and forming a resource crowding-out effect on GI [6]. A typical phenomenon is excessive donations by agents seeking personal social prestige or even using charitable donations as "political contributions" to seek political connections [5, 6]. Under the influence of such agent behavior, the business practice has given rise to the odd phenomenon that some companies actively seek GS while making "generous" donations for years on end. For example, through "generous" donations reigning the "China Charity Award" for many years, Kangmei Pharmaceuticals CO., LTD has received almost all GS in China's Guangdong Province. However, its R& D investment accounted for less than 3% yearly. Even in 2018, when it was exposed to "financial fraud," it still received government R& D subsidies of more than 26 million yuan, while its R& D investment that year was less than 1% of revenue. As a result, some scholars have questioned whether firms are willing to invest GS in innovation activities [7, 8].

The synthesis of the above views shows that most of the existing studies explored the relationship between the two from a single theoretical perspective. But in fact, while CG sends positive signals to the outside world to obtain external resource support, it also takes up internal resources and competes with GI for resources. Therefore, there is a complex nonlinear relationship between the two, and it is necessary to explore the relationship based on a more comprehensive theoretical perspective. In addition, in the Chinese institutional context, GS play an essential role in the relationship between CG and GI. On the one hand, since the current institutional environment for promoting GI in China is still imperfect, GS have become an essential external resource for enterprises to carry out GI [9], not only as a direct financial subsidy to reduce the cost of GI of enterprises but also as a clear signal to the outside world that subsidized enterprises receive government attention and reduce their financing constraints [10]. On the other hand, most enterprises do not passively accept GS but use some legal ways to build political connections to obtain more GS. Among the legal ways, CG, as an important way for companies to build and strengthen political-enterprise relations [11, 12], are

considered an essential strategy for companies to seek GS actively [5]. So, while GS can bring external resource support to enterprises, do they also induce an irrational allocation of internal resources? Is there an "optimal" CG interval that can realize the virtuous cycle of GS and charitable donations to promote GI?

To address the above issues, this paper has addressed the characteristics of enterprises' internal and external information asymmetry based on a research perspective that integrates signaling theory(ST) and principal-agent theory(PAT). It focused on external resource acquisition and internal resource allocation behaviors that affect enterprises' GI activities. Then we used CG as an entry point to conduct a two-step study in turn with the help of a threshold panel model. First, the input boundaries of CG that have different effects on GI are clarified. Then, on this basis, the relationship between GS and GI is explored under different levels of CG intensity. The aim is to develop instructive theoretical knowledge for improving the level of corporate GI and the rational allocation of resources by enterprises.

By doing so, we found the following: (1)CG show an inverted U-shaped threshold effect on GI, i.e., there is a weak correlation interval, a promotion interval and a crowding-out interval for CG on GI, respectively. (2) With the increase of CG, the promotion effect of GS on enterprise innovation investment shows a trend of diminishing marginal benefits, i.e., at different levels of CG, the impact of GS on GI shows "optimal" promotion effect interval, "suboptimal" promotion effect interval, "weak" promotion effect interval, respectively.

The contributions of this work are as follows: (1)This study integrated the theoretical perspectives of ST and PAT, which provided a new theoretical perspective for an in-depth study of GI.(2)The results of this study enriched the findings related to the relationship between CG and GI, which provided a specific theoretical basis for further exploring how enterprises can formulate scientific CG strategies under the trend of normalized CG.(3) The findings of this study provided a new paradigm for answering the controversial question of "whether firms invest GS in GI" in the study of the relationship between GS and GI, enriched the research literature in the field of GS, and provides a helpful reference for studying GS from a diversified perspective.

The remainder of this work is organized as follows. Section 2 briefly describes the literature that provide the basis for this work, construct a theoretical framework that integrates the perspectives of ST and PAT, and propose research hypothesis; Section 3 introduces the methodology used in the research; Section 4 demonstrates and analyze the results of the research. Robustness test was involved. Section 5 summarizes the main findings of the study and concludes with the managerial implications for further research.

## Literature review and hypothesis

### Corporate charitable giving and green innovation

**External innovation resource acquisition: A perspective of ST.**   Because of the limited internal resources, companies cannot carry out GI without the support of resources outside the organization and therefore have a strong incentive to establish good relationships with external stakeholders to obtain critical resources to enhance innovation capabilities [13]. The signaling effect of CG meets companies' needs to obtain key stakeholders' resources [3, 13], so CG can be given strategic importance to promote corporate GI. Based on the viewpoint of ST, CG can send positive signals to the outside world about the company's good financial capability and social responsibility, enhance the external resource owners' perception of the company's quality, alleviate the negative impact of information asymmetry between the two sides, and thus promote the acquisition of external innovative resources. Specifically, on the one hand, the implementation of charitable donations by enterprises can release signals to the

public that they are in good financial condition [3], especially in the context of the rapid development of green-finance and green-credit in China's capital market. On the other hand, Chinese government departments, as the allocators of some essential innovation resources, are an important source of external innovation resources for companies. CG can release to the government the corporate attitude of actively engaging in social responsibility and help enterprises gain more attention from the government, thus better-accessing resource support from the government (e.g., innovation project approval, innovation subsidies, policy support, and essential information resources), creating a resource effect that promotes corporate GI [9, 12, 14].

**Internal resource allocation: A perspective of PAT.** Under the PAT perspective, scholars have developed a view contrary to ST, which suggests that there is a crowding-out effect of CG on GI. Although PAT also agrees with the limited nature of resources within the firm, the theory focuses on the adverse effects of agency problems caused by information asymmetry on the allocation of firm resources, arguing that there is a potential conflict of interest between agents and principals and that this conflict can adversely affect the firm's innovative activities [15]. In implementing corporate CG, agents often generate or exacerbate excessive giving behaviors to pursue personal reputation and social status [16], thereby crowding out corporate GI resources. For example, Kangmei Pharmaceutical Co., Ltd. spends hundreds of millions of dollars annually on CG. At the same time, its annual innovation investment accounts for less than 1% of its revenue, which is far below the industry standard.

**Corporate charitable giving and green innovation in the integrated perspective.** ST emphasizes that CG can help companies send positive signals to critical external stakeholders in favor of corporate development to strengthen the relationship with key stakeholders and thus obtain external innovation resources to support the improvement of GI [3]. On the opposite hand, PAT argues that corporate agency problems can lead to excessive CG [15], which has a negative impact on GI by generating resource competition with corporate innovation activities. A review of the existing research findings shows that the reason for these two contradictory views lies in the difference in the focus of theoretical perspectives. The former takes the perspective of external resource acquisition, while the latter focuses on the internal allocation and utilization of resources, and the two theoretical perspectives are well complementary. Therefore, to address the complexity of the impact of CG on GI, this study argues that it is necessary to integrate ST and PAT to conduct a more comprehensive study on the impact mechanism of GI.

Overall, under the integrated ST-PAT perspective, there is a complex nonlinear relationship between corporate CG and GI, and the promoting and crowding out effects of CG on GI may coexist, and both of them are mutually reinforcing. First, when corporate CG is at a reasonable level, excessive CG has not yet arisen. The agency problem in corporate CG behavior has not arisen or is not severe. Within this reasonable level interval, the promoting effect of corporate CG on GI dominates. The reason is that, on the one hand, CG will also send a positive image signal to the public, thus attracting more external financial support to promote GI, such as BYD Co., Ltd., which made significant contributions through donations in the fight against the COVID-19 epidemic, not only reaped good social praise but also ushered in a stock "surge" and multi-party support after the epidemic, greatly accelerating the implementation of its DMI, blade batteries, and other new energy technologies. On the other hand, as the level of CG increases, it will play an increasing role in supporting the government's public service work, such as timely relief of the financial pressure encountered by the government in disaster relief, poverty alleviation and other work, which helps to build an excellent government-enterprise relationship. In return for reciprocal exchange, the government will give more "care" to such enterprises in allocating GS and other scientific and technological resources [14, 17, 18].

Second, when CG exceeds a reasonable level, corporate CG is in a state of over-giving, diverting many internal resources from the firm and competing with GI for resources [19]. At the same time, the agency problem will further aggravate the over-giving problem and form a crowding-out effect that inhibits GI [16]. In addition, under the influence of Chinese society's donation culture of administrative fundraising and public welfare apportionment, the good government-enterprise relationship between enterprises and local governments can also induce excessive corporate donations [20], exacerbating the crowding-out effect on GI. In summary, when the level of CG is in a reasonable range, the level of corporate GI will increase with the boost of CG investment. When charitable donations exceed a certain threshold, charitable donations will have a negative effect on corporate GI. The further increase of investment magnitude causes a crowding-out effect on GI.

Accordingly, we propose the following hypothesis:

H1: CG shows a positive and then negative inverted U-shaped threshold effect on GI.

## Government green subsidies, corporate charitable giving and green innovation

**Promotional role of GS: A perspective of ST.**    In the current Chinese market environment, the development of external resources such as financing environment, laws and regulations, and product markets related to GI is still in its initial stage, so GS are pretty scarce and essential for enterprises to carry out GI, and they are the essential external resources for enterprises to carry out GI [14, 16]. Therefore, GS can have a promoting effect on corporate GI. Specifically, first, GS have a scarcity feature in corporate GI. Due to the information inequality and the uncertainty characteristic of enterprise innovation activities, it is often difficult for the providers of funds external to the organization to judge the expected return on investment, making enterprise innovation activities often face financing constraints [21]. At the same time, the government has a very high authority in the Chinese context, and access to GS can directly send clear investment signals to the market about the company, thus helping the company obtain external financial support more easily [4]. Second, GS can play an essential role in promoting GI activities by enterprises. With direct financial support, GS can directly share the cost of innovation and thus increase firms' willingness to innovate. On the one hand, for enterprises that are not engaged in GI, GS can be used for the start-up of innovation activities such as equipment purchases to reduce the start-up costs of GI activities. On the other hand, for enterprises that have already started GI activities, GS can improve the marginal benefits of innovation activities by further reducing the marginal input costs of innovation, thus increasing the motivation of enterprises to GI [8].

**Weakening of the promoting effect: A perspective of PAT.**    As mentioned in the introduction, in the relationship between GS and corporate GI, the government plays the role of a principal, granting the subsidies to companies to encourage them to carry out GI. However, as agents of GS, companies tend to take advantage of information asymmetry to create agency problems [15]. Therefore, some scholars have questioned whether firms are willing to invest GS in GI activities [7, 8], and a series of studies have been conducted on this controversial but important topic. Specifically, first, due to information asymmetry, enterprises are prone to rent-seeking behaviors between government officials and enterprises in the process of obtaining GS, while GS may also induce distorted investment behaviors of enterprises and trigger excessive increases in raw materials to reduce marginal revenue, thus weakening enterprises' willingness to GI [22]. Second, the negative impact of firms' behavior of capturing GS on GI exists. The existence of loopholes in the system [23] or inadequate regulation makes it possible

for firms to engage in strategic behaviors to capture and cheat the government's subsidies or to adopt low-quality "strategic innovations" to obtain higher short-term returns [2]. For example, in the process of the Chinese government subsidizing the development of the new energy vehicle industry, the "fraudulent subsidy gate" incident occurred, in which the companies involved cheated a large number of GS through false reports and other methods, which seriously disrupted the healthy development of the new energy vehicle industry.

**GS, CG and GI in the integrated perspective.** A review of the above literature revealed that the former, based on the ST, focused on external resource acquisition and argued that GS reduce the cost of GI inputs for enterprises while releasing good signals to society for enterprises to help reduce financing constraints, which has a promoting effect on GI. The latter considered the principal-agent problem and argues that firms' internal allocation and utilization of resources affect the utilization of GS. In addition, previous studies have not considered the critical role of CG in the relationship between the two. However, the strategic function of CG in business practice to build the relationship between government and business should not be ignored [2]. In this regard, this study takes CG as an entry point and argues that corporate CG plays a critical trade-off role in the relationship between the two. Specifically, firstly, when enterprises start to make charitable donations, the relationship between government and enterprises will be strengthened with the improvement of their charitable donations. Enterprises can obtain more innovative resources from the government [24]. At this time, charitable donations and GS form a synergistic effect of jointly promoting GI. GS are in the "strong promotion level" for GI. However, when CG exceeds a specific limit, it will weaken the promotion effect of GS. The reason is that, on the one hand, the scale of GS is limited by government policies, and there is an upper limit of subsidies for enterprises, so too much charitable donations will crowd out enterprises' innovation resources and reduce the promotion effect of GS on enterprises' GI. On the other hand, when enterprises over-invest in charitable donations, they tend to be caught in complex interactions with stakeholders [25], such as excessive maintenance of government-enterprise relations, which creates a resource curse and reduces the incentive to innovate [16]. In summary, when the CG input is at a reasonable level, GS positively promote GI as the input intensity increases; when CG exceeds a specific limit, the promoting effect of GS on corporate GI will gradually diminish. Therefore, we propose the following research hypothesis.

H2: There is a threshold effect of GS on GI in the presence of differences in CG. With increased CG investment, GS show a promotion effect of diminishing marginal benefit on GI.

## Research methodology and data description

### Sample construction

This study takes manufacturing listed companies in China from 2016 to 2019 (For the following two reasons, the sample in this study was selected with 2016 as the initial year of the research sample. One is that in 2014, the General Office of the State Council of China issued the Action Plan for Energy Conservation, Emission Reduction and Low Carbon Development 2014–2015, which officially kicked off the green energy conservation and emission reduction activities in various industries across China, and green innovation activities were officially launched in China in full scale after 2015. Second, by reviewing the annual reports of listed companies, we found that since 2015, the annual reports of listed companies began to explicitly disclose the types of government subsidies received, which helped to more accurately measure relevant variables such as government subsidies.) as the research sample and does the following. (1) Exclude the sample of listed companies in the ST category with abnormal and missing values in the data during the study period; (2) In order to eliminate the influence of extreme

values on the overall smoothness of the data, this paper performs a Winsorize tailing process of 1% above and below for continuous variables. After the balanced processing of the sample, 863 listed companies with balanced panel data from 2016 to 2019 were obtained, totaling 3452 sample data. Among them, the data related to green patents are based on the green patent international patent classification (IPC) list issued by the World Intellectual Property Organization and obtained by hand through the State Intellectual Property Office (SIPO) website for manual collation. GS, corporate CG, and other related statistics are obtained from China Stock Market & Accounting Research Database (CSMAR). Some missing data are obtained manually from the annual reports of listed companies. The data are processed by statistical analysis software such as EXCEL2016 and STATA14.0.

## Variable measurements

**Corporate green innovation (GI).** Considering the complex procedure of patent approval and other uncontrollable factors, the approval of some patents often takes many years, and it is difficult to control the time lag and other issues. It is impossible to guarantee the relevance of a certain period of GS and charitable donations to its relationship. Therefore, this study considers that the number of green patents filed by a company is a better picture of the company's GI activities in that year. According to the definition given by WIPO, green patents refer to patents granted for "green" technologies in new energy such as solar energy, hybrid vehicles, wind energy, fuel cell vehicles, tidal, geothermal, biofuel, and carbon capture and storage and nuclear energy, etc. Compared with general patents, green patents have significant environmental friendliness and resource conservation advantages. Green patents have significant advantages over general patents regarding environmental friendliness and resource conservation. According to China's Patent Law, patents include invention patents, utility model patents, and design patents, of which invention patents refer to new technical solutions for products, methods, or improvements thereof. And the utility model patents refer to new technical solutions for the shape, structure or combination of products suitable for practical use; and design patents refer to new designs for the shape, pattern or combination of products and the combination of color and shape and pattern that are aesthetically pleasing and suitable for industrial application. It is now generally accepted in the academic community that invention patents and utility model patents are more advantageous in terms of technical importance, economic value, and degree of innovation and can accurately measure the output and quality of corporate innovation activities [26]. Therefore, this study adopts the sum of green invention-type and utility-type patent applications to measure the level of GI of enterprises with reference to the study of Wang Xin& Wang Ying [27]and treats it logarithmically in the regression model. In addition, considering that there may be a tendency to allocate GS, and there is also a lag effect of corporate charitable donations on innovation inputs, in order to avoid the endogeneity problem arising from reverse causality, this study treats GI as t+1 period, i.e., $GI_{t+1}$ = LN (the number of green invention-type patent applications $_{t+1}$ + the number of utility model patent applications $_{t+1}$+1).

**Charitable giving (CG).** In order to reduce the impact of corporate size on the amount of corporate CG, this study chose to measure the level of corporate CG by relative donation amount [28], i.e., CG = (annual social donation amount/business revenue of the year) * 100.

**Government green subsidies (GS).** Drawing on Xinle's study [25], GS = (amount of direct government green subsidies /enterprise operating income) * 1000. Where the amount of direct government green subsidies draws on Hu's study [29] on the measurement of GS, the number of government green subsidies for GI disclosed in the annual reports of listed companies is used as a proxy variable for the amount of direct GS.

**Table 1. The measurements of all variables.**

| Name of variables | Symbols | Definition or measurement |
|---|---|---|
| Green Innovation$_{t+1}$ | GI$_{t+1}$ | LN (the number of green invention-type patent applications $_{t+1}$ + the number of utility model patent applications $_{t+1}$+1) |
| Charitable Giving | CG | (annual social donation amount/business revenue of the year) * 100 |
| Government Green Subsidies | GS | (amount of direct government green subsidies/enterprise operating income) * 1000 |
| firm profitability | ROA | Net income / Total assets at the end of the period |
| Tobin's Q | TBQ | Year-end market capitalization / Year-end total assets |
| Gearing ratio | LEV | Gearing ratio |
| Fixed assets ratio | FT | Ending fixed assets / Ending total assets |
| Industry Competitiveness | HHI | 1—Herfindahl Index |
| firm scale | Scale | LN(1 + number of employees in the company) |

**Control variables (Controls).** Drawing on the existing literature by scholars Zhang W [30] and Yang [24], the following variables are mainly controlled for: firm profitability (ROA), Tobin's Q (TBQ), gearing ratio (LEV), fixed asset ratio (FT), industry competitiveness (HHI), and firm Scale (Scale), which are measured in Table 1.

**Design of the model.** Based on the above analysis, this study used the fixed-threshold panel data model proposed by Hansen [31] to construct a triple-threshold panel model by automatically identifying the data through the Bootstrap method to determine the threshold values. Model (1) is a multiple threshold model with corporate charitable giving (CG) as both the core explanatory variable and the threshold variable, and green innovation (GIt+1) as the explanatory variable. Model (2) is a multiple threshold model with government green subsidies (GS) as the core explanatory variable, corporate charitable giving (CG) as the threshold variable, and green innovation (GIt+1) as the explanatory variable.

$$
\begin{aligned}
GI_{i,t+1} = {} & u_i + \alpha_1 CG_{i,t}I\big(CG_{i,t} \leq \gamma_1\big) + \alpha_2 CG_{i,t}I\big(\gamma_1 < CG_{i,t} \leq \gamma_2\big) + \alpha_3 CG_{i,t}I\big(\gamma_2 < CG_{i,t} \leq \gamma_3\big) \\
& + \alpha_4 CG_{i,t}I\big(\gamma_3 < CG_{i,t}\big) + \alpha_{5i}\sum Controls_{i,t} + e_{i,t}
\end{aligned}
\tag{1}
$$

$$
\begin{aligned}
GI_{i,t+1} = {} & u_i + \alpha_1 GS_{i,t}I\big(CG_{i,t} \leq \gamma_1\big) + \alpha_2 GS_{i,t}I\big(\gamma_1 < CG_{i,t} \leq \gamma_2\big) + \alpha_3 GS_{i,t}I\big(\gamma_2 < CG_{i,t} \leq \gamma_3\big) \\
& + \alpha_4 GS_{i,t}I\big(\gamma_3 < CG_{i,t}\big) + \alpha_{5i}\sum Controls_{i,t} + e_{i,t}
\end{aligned}
\tag{2}
$$

Where the subscripts $i$ = 1,2,3. . ., $t$ = 1,2,3. . .; $GI_{t+1}$ represents corporate green innovation in period t+1; CG represents corporate charitable giving; $I(\cdot)$ is an indicator function that takes the value of 1 when the condition in parentheses holds, and 0 otherwise; $\gamma_i$ is the threshold value. When $\gamma_1 = \gamma_2 = \gamma_3$, the model is a single threshold model. The model is a double threshold model when $\gamma_1 \neq \gamma_2 = \gamma_3$. When $\gamma_1 \neq \gamma_2 \neq \gamma_3$, the model is a triple threshold model; *Controls* are the control variables; $u_i$ denotes individual fixed effects to reflect the inter-individual differences in the sample, and $e_{i,t}$ is the random error term.

## Empirical results

### Descriptive statistics

The descriptive statistics and correlation analysis of each variable are shown in Table 2. In the sample data, the dependent variable GI$_{t+1}$ has a mean value of 1.523, a standard deviation of 0.758, and a minimum value of 0, which meets the sufficient necessary conditions for a normal distribution, thus initially indicating the applicability of this regression model selection. The

**Table 2. Descriptive statistics and correlations.**

| variables | 1 | 2 | 3 | 4 | 5 | 6 | 7 | 8 | 9 |
|---|---|---|---|---|---|---|---|---|---|
| 1 $GI_{t+1}$ | 1.000 | | | | | | | | |
| 2 CG | -0.289*** | 1.000 | | | | | | | |
| 3 GS | 0.036** | 0.183*** | 1.000 | | | | | | |
| 4 ROA | -0.004* | -0.067*** | 0.060*** | 1.000 | | | | | |
| 5 TBQ | 0.290*** | -0.327*** | -0.027* | 0.141*** | 1.000 | | | | |
| 6 LEV | -0.318*** | 0.453*** | 0.126*** | -0.246*** | -0.380*** | 1.000 | | | |
| 7 FT | -0.257*** | 0.223*** | 0.071*** | -0.031* | -0.158*** | 0.166*** | 1.000 | | |
| 8 HHI | -0.143*** | 0.027* | 0.025 | -0.058*** | -0.088*** | 0.042** | -0.012 | 1.000 | |
| 9 Scale | -0.312*** | 0.421*** | 0.217*** | 0.034** | -0.404*** | 0.479*** | 0.183*** | 0.047*** | 1.000 |
| Mea | 1.223 | 0.801 | 1.277 | 0.033 | 2.314 | 0.408 | 0.202 | 0.196 | 7.858 |
| Std. Dev | 0.758 | 9.438 | 3.014 | 0.085 | 1.441 | 0.197 | 0.150 | 0.138 | 1.308 |
| Min | 0.000 | 0.000 | 0.000 | -1.068 | 0.871 | 0.055 | 0.000 | 0.036 | 4.174 |
| Max | 4.434 | 470.179 | 74.373 | 0.379 | 8.464 | 0.887 | 0.876 | 1.000 | 12.621 |
| N | 3452 | 3452 | 3452 | 3452 | 3452 | 3452 | 3452 | 3452 | 3452 |

***, **, * indicate at P<0.01, P<0.05, P<0.1, respectively

minimum value of CG is 0, the maximum value is 470.179%, the mean value is 0.801%, the maximum value is about 586 times the mean value, and the standard deviation is 9.438%, which indicates that there is a large gap in the level of CG among the sample companies. The minimum value of GS is 0, the maximum value is 74.373%, the mean value is 1.277%, the maximum value is about 58 times the mean value, and the standard deviation is 3.014%, which indicates that there is also a large gap in the intensity of GS among the sample companies, but the overall gap is smaller than the level of CG gap. From the results of correlation analysis among variables, $GI_{t+1}$ was significantly negatively correlated with CG, ROA, TBQ, LEV, HHI, and Scale; $GI_{t+1}$ was significantly positively correlated with GS and TBQ. In addition, the results in Table 2 show that the absolute values of the correlation coefficients among the variables are below 0.5, indicating that there is no severe problem of multicollinearity among the independent variables in this study.

### Threshold variable estimation

The results of the threshold effect test are shown in Table 3: The double threshold effects in both model (1) and model (2) are significant at the 1% level and do not pass the triple threshold effect test, so the double threshold model is used for analysis in both model (1) and model

**Table 3. Self—sampling test of the threshold effect.**

| Models | Threshold effects | F | P | Bootstrap times | The critical value | | |
|---|---|---|---|---|---|---|---|
| | | | | | 1% | 5% | 10% |
| Model (1) | Single threshold | 14.67** | 0.020 | 300 | 14.331 | 15.994 | 19.207 |
| | Double threshold | 21.27*** | ≤0.001 | 300 | 18.846 | 26.809 | 30.766 |
| | Triple threshold | 3.12 | 0.900 | 300 | 22.14 | 34.498 | 77.873 |
| Model (2) | Single threshold | 36.96*** | ≤0.001 | 300 | 11.956 | 14.962 | 21.730 |
| | Double threshold | 83.64*** | ≤0.001 | 300 | 8.699 | 9.171 | 22.784 |
| | Triple threshold | 8.92 | 0.700 | 300 | 18.978 | 35.190 | 60.5619 |

***, **, * denote P<0.01, P<0.05, and P<0.1, respectively. The following table is the same.

**Table 4. Estimates and confidence intervals for the threshold.**

| Models | | Threshold value | 95% Confidence interval |
|---|---|---|---|
| Model (1) | $\gamma_1$ | 0.645 | [0.618,0.691] |
| | $\gamma_2$ | 0.758 | [0.741,0.765] |
| Model (2) | $\gamma_1$ | 1.611 | [1.396,1.630] |
| | $\gamma_2$ | 1.932 | [1.911,2.112] |

(2). Table 4 reports the threshold estimates and confidence intervals of models (1) and (2), respectively. The results of model (1) indicate that in the relationship between CG and GI, the threshold thresholds for CG are 0.645 and 0.760, and the confidence intervals are [0.618,0.691] and [0.741,0.765], respectively; the results of model (2) indicate that in the relationship between GS and GI, the threshold thresholds for CG are 1.620 and 1.921, respectively. The confidence intervals are [1.396,1.630], [1.911,2.112], respectively. All the above thresholds passed the authenticity test.

## Analysis of threshold regression results

According to the regression results of model (1) in Table 5, when the CG of the firm is at low input intensity (CG≤0.645), although the coefficient between CG and $GI_{t+1}$ is negative, it is not significant, probably because the CG input level is low and does not have a substantial effect on $GI_{t+1}$; when the CG is at medium input intensity (0.645 < CG≤ 0.758), its impact coefficient is 1.645 (P<0.01), indicating that there is a significant positive contribution of CG to $GI_{t+1}$ as CG increases; when CG is at a high input level (0.758< CG), its impact coefficient is -0.082 (P<0.05), indicating that when CG grant exceeds a certain limit, it will have a

**Table 5. Regression results.**

| Variables | Model (1) | | | Model (2) | | |
|---|---|---|---|---|---|---|
| | Coefficient estimates | P-Value | T-Value | Coefficient estimates | P-Value | T-Value |
| ROA | 0.863 | 0.215 | 1.241 | 0.853 | 0.205 | 1.271 |
| TBQ | -0.311*** | ≤0.001 | -6.182 | -0.354*** | ≤0.001 | -7.182 |
| LEV | -0.796 | 0.266 | -1.131 | -0.872 | 0.23 | -1.271 |
| FT | 1.181 | 0.241 | 1.173 | 1.632* | 0.096 | 1.661 |
| HHI | -0.457 | 0.742 | -0.331 | -1.089 | 0.420 | -0.810 |
| Scale | 0.275 | 0.158 | 1.410 | 0.352* | 0.062 | 1.870 |
| CG_1 | -0.142 | 0.680 | -0.411 | | | |
| CG_2 | 1.645*** | ≤0.001 | 4.632 | | | |
| CG_3 | -0.082** | 0.041 | -1.661 | | | |
| GS _1 | | | | 0.069** | 0.036 | 2.100 |
| GS _2 | | | | 0.059*** | ≤0.001 | 13.210 |
| GS _3 | | | | 0.006*** | ≤0.001 | 6.830 |
| N | 3,452 | | | 3,452 | | |
| R² | 0.030 | | | 0.085 | | |
| F-value | 8.100*** | | | 26.600*** | | |

a. CG _1, CG _2, and CG _3 are the low, medium, and high charitable giving intensity intervals, respectively; GS _1, GS _2, and GS _3 are the low, medium, and high charitable giving intensity intervals, respectively; b.T is the t-value of the coefficient significance test under the condition of heteroskedasticity; c.P-value is the result of using "Bootstrap" to obtain 300 times repeated sampling; d."***, **, *" denote P<0.01, P<0.05, and P<0.1, respectively. e.P value is the result obtained by using "Bootstrap" to repeatedly sample 300 times; f."***, **, *" indicate P<0.01, P<0.05, and P<0.1, respectively. The following table is the same

significant negative effect on $GI_{t+1}$. In summary, CG shows a positive and then negative inverted U-shaped threshold effect on GI, thus the hypothesis H1 was verified.

By observing the variation of the coefficient between CG and GI it can be found that: The effect of CG on GI is as the saying "too much water drowned the miller" goes, i. e. too much CG will have a negative effect on GI. This result is consistent with the view of scholar Wang [12].CG can send positive signals to the outside world such as being financially sound or socially responsible, thus helping firms to obtain more external resource support [13]. However, if the level of CG input is low, it neither draws external attention to obtain external resource support nor takes up too much of the organization's internal resources, so it does not have a substantial impact on the firm's GI. When the level of CG input is too high, CG will overly occupy the internal resources of the organization to have a negative impact on GI. Therefore, a reasonable level of CG input is what will promote GI, and the management of CG should be emphasized at the strategic level of the organization.

According to the regression results of model (2) in Table 5, the coefficient between GS and $GI_{t+1}$ is 0.069 (P<0.05) when CG is at low input intensity (CG≤1.611), indicating that there is a significant positive effect of GS on $GI_{t+1}$; when CG is at medium input intensity (1.611< CG≤1.932), its impact coefficient decreases to 0.059 (P<0.01), indicating that the positive effect of CG on $GI_{t+1}$ decreases; when CG is at high input level (1.932< CG), its impact coefficient decreases to 0.006 (P<0.01), and the positive effect of GS on corporate $GI_{t+1}$ has a more significant weakening. The above results suggest a marginal decreasing promotion effect of GS on GI as the intensity of corporate CG increases. In summary, there is a threshold effect of GS on GI in the presence of differences in CG. With increased CG investment, GS show a promotion effect of diminishing marginal benefit on GI. Thus the hypothesis H2 is verified.

By observing the variation of the coefficient between GS and GI it can be found that: There was a positive contribution of GS to GI at different levels of CG input intensity. The result is consistent with scholar Bai's view [9] that GS promotes GI. In addition, the promotion effect of GS on GI is continuously weakened as the intensity of CG input increases, i.e., there is an optimal promotion interval of GS on GI under different CG input levels.

## Robustness test

Based on practice and existing scholarly research, equity structure is also an essential factor influencing firms to carry out GI [32]. Drawing on the test method of adding control variables to the threshold effect model by Liu Lanjian et al. [33], the shareholding ratio of the first largest shareholder(RFS) is used to measure the equity structure, which is included as an additional control variable in the above model for robustness testing. The test results are shown in Table 6, and the direction of the regression coefficients of the test results is approximately the same as the total sample regression results. The significance also remains the same, indicating that the results of this paper have good robustness and reliability.

## Discussion and conclusion
### Conclusions

Based on the integrated perspective of resource dependence theory and principal-agent theory, this paper focuses on the impact of corporate CG and GS on GI, and explores the impact of corporate CG on GI and the impact of GS on GI under the threshold of CG with the help of threshold regression models, respectively, using domestic manufacturing listed companies from 2016 to 2019, and draws the following conclusions.

(1) CG show an inverted U-shaped threshold effect on GI, i.e., there is a weak correlation interval (CG≤0.645), a promotion interval (0.645< CG≤0.758), and a crowding-out interval

**Table 6. Robustness test.**

| Variables | Model (1) | | | Model (2) | | |
|---|---|---|---|---|---|---|
| | Coefficient estimates | P-Value | Variables | Coefficient estimates | P-Value | Variables |
| ROA | 0.146 | 0.454 | 0.751 | 0.285 | 0.125 | 1.483 |
| TBQ | -0.050*** | 0.002 | -3.112 | -0.049*** | $\leq$0.001 | -3.084 |
| LEV | -0.391* | 0.063 | -1.861 | -0.386* | 0.07 | -1.831 |
| FT | 0.345 | 0.235 | 1.194 | 0.212 | 0.396 | 0.821 |
| HHI | -0.172 | 0.709 | -0.371 | -0.095 | 0.320 | -0.210 |
| Scale | 0.255 | 0.158 | 0.482 | 0.049* | 0.051 | 0.841 |
| RFS | 0.006 | 0.174 | 1.361 | 0.006 | 0.105 | 1.443 |
| CG_1 | -0.077 | 0.632 | 1.55 | | | |
| CG_2 | 0.193** | 0.022 | 2.03 | | | |
| CG_3 | -0.021*** | $\leq$0.001 | -4.23 | | | |
| GS _1 | | | | 0.041** | 0.024 | 3.71 |
| GS _2 | | | | 0.009*** | $\leq$0.001 | 2.71 |
| GS _3 | | | | 0.001** | $\leq$0.02 | 1.83 |
| N | 3,452 | | | 3,452 | | |
| R² | 0.021 | | | 0.017 | | |
| F-value | 5.035*** | | | 7.798*** | | |

(0.758< CG) for CG on GI, respectively. From the results, 32.59% of the sample firms are located in the weak correlation interval, i.e., CG neither promotes nor hinders GI in this interval; only 4.896% of the sample firms are located in the promotion interval; 62.514% of the sample firms are located in the crowding-out interval. That indicates the majority of the sample companies (62.514%) have a level of CG that is not conducive to the improvement of GI, and only a small number of sample companies (4.896%) have achieved synergistic development CG and GI.

(2) With the increase of CG, the promotion effect of GS on enterprise innovation investment shows a trend of diminishing marginal benefits, i.e., at different levels of CG, the impact of GS on GI shows "optimal" promotion effect interval (CG≤1.611), "suboptimal" promotion effect interval (1.611< CG≤1.932), "weak" promotion effect interval (1.932< CG). From the results, 59.733% of the sample data were located in the "optimal" promotion interval, 4.867% in the "suboptimal" promotion interval, and 35.380% in the "weak" promotion interval. The results indicate that there is a clear "polarizing" in the relationship between GS and GI, i.e., the majority of enterprises (59.733%) are in the "optimal" promotion zone of GS and GI, but a significant number of enterprises (35.380%) are in the "weak" promotion zone, resulting in the waste of GS.

## Theoretical contributions

① Integrating ST and PAT provides a new research perspective for GI. This study finds through literature combing that corporate GI not only depends on the support of external resources but also is profoundly affected by internal resource allocation. Based on the integrated ST and PAT perspectives, this study explores the nonlinear relationships between CG and GS on GI, respectively, providing a more comprehensive research idea to study the influence mechanisms affecting corporate GI.

② The findings related to the relationship between CG and GI are enriched. The findings of this study verify that corporate CG presents a positive and then negative inverted U-shaped threshold effect on GI and are based on confirming the views of scholars such as Shen [19] and

WANG [34] on the existence of promoting and inhibiting effects of CG on corporate innovation activities, the input boundary of the promoting and inhibiting effects of corporate CG on GI is further clarified, which provides a specific theoretical basis for the normalization of corporate CG. That provides a specific theoretical basis for further exploring how enterprises can formulate scientific CG strategies under the trend of normalized corporate CG.

③ The research paradigm of GS on GI is extended. Using CG as the entry point, a threshold regression model confirms that the promotion effect of GS on GI investment is influenced by the level of CG of firms, and the promotion effect of GS on firms' innovation investment tends to diminish in marginal benefit as CG increase. That provides a new paradigm for answering the controversial question of "whether firms invest GS in GI" in the study of the relationship between GS and GI [7, 8], enriches the research literature in the field of GS, and provides a helpful reference for studying GS from a diversified perspective.

## Practical implications

This study explores the non-linear relationship between corporate CG and GS on GI, which has specific reference significance for enterprises and government departments to promote corporate innovation activities. Firstly, enterprises should pay attention to the planning and management of charitable donations from the corporate strategy level. Reasonable donations can fulfill corporate social responsibility and obtain more external resources support so that enterprises can establish competitive advantages in GI and achieve economic and social goals. Secondly, enterprises should develop a scientific and transparent monitoring mechanism to reduce the negative impact of agency problems on enterprise resources, coordinate the dynamic balance between corporate philanthropic strategy and R&D strategy, effectively play the role of GS to promote corporate GI and help China achieve the "double carbon" strategic goal with high quality.

For the government, it should play the dual role of manager of GS and supervisor of enterprises' CG behavior. From the findings of this paper, we can find that GS can help enterprises improve their GI level. However, excessive donations from enterprises can weaken the promotion effect of GS. Therefore, on the one hand, the government should recognize the positive effect of GS on enterprises' GI, continuously improve the audit and management mechanism of subsidies, and strengthen the control of enterprises' strategic innovation and other "speculative" behaviors. At the same time, they should be wary of ineffective subsidies caused by inappropriate government-enterprise relations. On the other hand, we should give full play to the guidance and service role of the market economy, strengthen the guidance of corporate CG behavior, encourage responsible market activities and commercial behavior by improving the market mechanism, build good support and guidance measures, guide the transformation of excessive CG to scientific CG, and effectively eliminate the "polarizing " phenomenon of CG among sample enterprises. "The study also aims to scientifically promote the sustainable development of corporate philanthropy and corporate social responsibility.

## Limitations and future research

This study has some shortcomings and needs further improvement. First, this study only obtained data related to domestic listed companies from 2016 to 2019 for regression analysis, and the generalizability of the research findings needs further verification. At the same time, the study sample is all listed companies, which reduces the study's validity to a certain extent. Therefore, in the subsequent study, we will add non-listed companies to the sample and further expand the period of the study sample to make further research judgments. Secondly, different ownership natures of enterprises have distinct differences in resource allocation due to

the limitation of space. This study did not further explore the nature of ownership, so it is necessary to conduct in-depth research in future studies. Finally, this study has only explored the relationship between charitable donations and GS on GI from the logic of resource acquisition and expenditure in a superficial way. Its theoretical contribution is limited, so it is necessary to further explore the theoretical depth in future studies.

## Author Contributions

**Conceptualization:** Hongpeng Wang.

**Data curation:** Hongpeng Wang.

**Formal analysis:** Hongpeng Wang.

**Funding acquisition:** Hongpeng Wang.

**Methodology:** Hongpeng Wang.

**Resources:** Hongpeng Wang.

**Software:** Hongpeng Wang.

**Supervision:** Hongpeng Wang.

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
