## [Decision Letter · Decision Letter 0]

26 Jul 2022

PONE-D-22-18951Does charitable giving reduce firms' willingness to invest in green innovation? --Based on the Integration Perspective of Signaling Theory and Agency TheoryPLOS ONE

Dear Dr. Wang,

Thank you for submitting your manuscript to PLOS ONE. After careful consideration, we feel that it has merit but does not fully meet PLOS ONE’s publication criteria as it currently stands. Therefore, we invite you to submit a revised version of the manuscript that addresses the points raised during the review process. Please submit your revised manuscript by Sep 09 2022 11:59PM. If you will need more time than this to complete your revisions, please reply to this message or contact the journal office at plosone@plos.org. Please include the following items when submitting your revised manuscript:A rebuttal letter that responds to each point raised by the academic editor and reviewer(s). You should upload this letter as a separate file labeled 'Response to Reviewers'.A marked-up copy of your manuscript that highlights changes made to the original version. You should upload this as a separate file labeled 'Revised Manuscript with Track Changes'.An unmarked version of your revised paper without tracked changes. You should upload this as a separate file labeled 'Manuscript'.

We look forward to receiving your revised manuscript.

Kind regards,

Ming Zhang, Ph.D.

Academic Editor

PLOS ONE

Journal Requirements:

Reviewers' comments:

Reviewer's Responses to Questions

**Comments to the Author**

1. Is the manuscript technically sound, and do the data support the conclusions?

Reviewer #1: Yes

2. Has the statistical analysis been performed appropriately and rigorously? 

Reviewer #1: Yes

3. Have the authors made all data underlying the findings in their manuscript fully available?

Reviewer #1: Yes

4. Is the manuscript presented in an intelligible fashion and written in standard English?

Reviewer #1: Yes

5. Review Comments to the Author

Reviewer #1: This study explores the impact of charitable giving on firms' willingness and green innovation. I think this paper is interesting but it is not well written. I think it should conduct a major revision before accepted. Some modifications should be conducted as follows:

1. The title can revise to “Does charitable giving reduce firms' willingness to invest in green innovation?”.

2. The abstract section is not well written. Authors should simply give the purpose of this paper, and mainly focus on the main findings of this paper.

3. The introduction section does not clearly give the main contributions of this paper, and the differences with the other literature.

4. Authors should summary literature and find out the knowledge gaps, instead of listing literature. Besides, the authors have cited a large number of papers from Chinese journals, I suggest authors should cite more papers from the famous journals, such as energy economics, JEEM etc. These related papers that you can refer to:

(1) Zhang, W.K., Luo, Q., Liu, S.Y. 2022. Is government regulation a push for corporate environmental performance? evidence from China. Economic Analysis and Policy. 74, 105-121.

(2) Wen, H., Lee, C.C., Zhou, F. (2022). How does fiscal policy uncertainty affect corporate innovation investment? Evidence from China’s new energy industry. Energy Economics. 105, 105767

(3) Dai, L., Mu, X., Lee, C. C., Liu, W. 2021. The impact of outward foreign direct investment on green innovation: the threshold effect of environmental regulation. Environ Sci. Pollut. Res.

5. The format of some references is not right, and I suggest authors should revise them, such as LUAN Qiang, Yang Yang, YU Xiaoyu, Hu D. I can find more than ten formats from the reference section.

6. In empirical application section, authors should give some discussion and analysis, rather than a brief report of the results.

7. Authors needs to check the full text.

6. PLOS authors have the option to publish the peer review history of their article (what does this mean?). If published, this will include your full peer review and any attached files.

Reviewer #1: No

---

## [Author Response · Author response to Decision Letter 0]

24 Aug 2022

Dear Dr. Zhang and reviewer：

Thank you for your letter and the reviewers’ comments on our manuscript entitled "Does charitable giving reduce firms' willingness to invest in green innovation? --Based on the Integration Perspective of Signaling Theory and Agency Theory" (Manuscript Number：PONE-D-22-18951). Those comments are very helpful for revising and improving our paper, as well as the important guiding significance to the future research. We have studied the comments carefully and made corrections which we hope meet with approval. The main corrections are in the manuscript and the responds to the reviewers’ comments are as follows (the replies are highlighted in blue).

Replies to the reviewers’ comments:

Reviewer :

1. The title can revise to “Does charitable giving reduce firms' willingness to invest in green innovation?”.

Response：We have revised the original title to “Does charitable giving reduce firms' willingness to invest in green innovation?”.

2. The abstract section is not well written. Authors should simply give the purpose of this paper, and mainly focus on the main findings of this paper. �

Response：We have added the purpose and main findings of the study in the abstract section. The additional contents are as follows：

···The purpose of this study is to clarify the mechanism of CG and government green subsidies(GS) on green innovation(GI). In this regard, we integrated signaling theory and principal-agent theory to provide a new theoretical perspective for simultaneously focus on the impact of external resource acquisition and internal resource allocation on GI）···Our findings indicate that there is an inverted U-shaped threshold effect of CG on GI. The impact of GS on GI shows a decreasing marginal benefit as the intensity of CG increases···

See L.40-51 of the manuscript（Revised Manuscript with Track Change）for details.

3. The introduction section does not clearly give the main contributions of this paper, and the differences with the other literature.

Response：In response to the comment, We have added the main contributions of this study in the introduction section to show the differences with other literature. The additional contents are as follows：

···By doing so, we found the following: (1)CG show an inverted U-shaped threshold effect on GI, i.e., there is a weak correlation interval, a promotion interval and a crowding-out interval for CG on GI, respectively. (2) With the increase of CG, the promotion effect of GS on enterprise innovation investment shows a trend of diminishing marginal benefits, i.e., at different levels of CG, the impact of GS on GI shows "optimal" promotion effect interval, "suboptimal" promotion effect interval, "weak" promotion effect interval, respectively.

The contributions of this work are as follows: (1)This study integrated the theoretical perspectives of ST and PAT, which provided a new theoretical perspective for an in-depth study of GI.(2)The results of this study enriched the findings related to the relationship between CG and GI, which provided a specific theoretical basis for further exploring how enterprises can formulate scientific CG strategies under the trend of normalized CG.(3) The findings of this study provided a new paradigm for answering the controversial question of "whether firms invest GS in GI" in the study of the relationship between GS and GI, enriched the research literature in the field of GS, and provides a helpful reference for studying GS from a diversified perspective.

The remainder of this work is organized as follows. Section 2 briefly describes the literature that provide the basis for this work, construct a theoretical framework that integrates the perspectives of ST and PAT, and propose research hypothesis; Section 3 introduces the methodology used in the research; Section 4 demonstrates and analyze the results of the research. Robustness test was involved. Section 5 summarizes the main findings of the study and concludes with the managerial implications for further research.

See L.137-160 of the manuscript（Revised Manuscript with Track Change）for details.

4.Authors should summary literature and find out the knowledge gaps, instead of listing literature. Besides, the authors have cited a large number of papers from Chinese journals, I suggest authors should cite more papers from the famous journals, such as energy economics, JEEM etc. These related papers that you can refer to:

(1) Zhang, W.K., Luo, Q., Liu, S.Y. 2022. Is government regulation a push for corporate environmental performance? evidence from China. Economic Analysis and Policy. 74, 105-121.

Zhang W, Luo Q, Liu S. Is government regulation a push for corporate environmental performance? Evidence from China[J]. Economic Analysis and Policy, 2022, 74: 105-121. 

(2) Wen, H., Lee, C.C., Zhou, F. (2022). How does fiscal policy uncertainty affect corporate innovation investment? Evidence from China’s new energy industry. Energy Economics. 105, 105767

(3) Dai, L., Mu, X., Lee, C. C., Liu, W. 2021. The impact of outward foreign direct investment on green innovation: the threshold effect of environmental regulation. Environ Sci. Pollut. Res.

Response：We are very grateful to the reviewers for this comment, which has provided great help to revise this manuscript. Firstly, we have added the knowledge gaps in the introduction section. The additional contents are as follows：

···The synthesis of the above views shows that most of the existing studies explored the relationship between the two from a single theoretical perspective. But in fact, while CG sends positive signals to the outside world to obtain external resource support, it also takes up internal resources and competes with GI for resources. Therefore, there is a complex nonlinear relationship between the two, and it is necessary to explore the relationship based on a more comprehensive theoretical perspective···

Secondly, we updated the other English literature based on the supplementation of the above 3 literature in order to minimize the original Chinese literature used to enhance the persuasive power of the views. See L.606-709 of the manuscript（Revised Manuscript with Track Change）for details.

5. The format of some references is not right, and I suggest authors should revise them, such as LUAN Qiang, Yang Yang, YU Xiaoyu, Hu D. I can find more than ten formats from the reference section.

Response：We have changed the format of all references to Plosone style. For example, Dai L, Mu X, Lee CC. et al. The impact of outward foreign direct investment on green innovation: the threshold effect of environmental regulation. Environ Sci Pollut Res. 2021; 28: 34868–34884. https://doi.org/10.1007/s11356-021-12930-w. See L.606-709 of the manuscript（Revised Manuscript with Track Change）for details.

6. In empirical application section, authors should give some discussion and analysis, rather than a brief report of the results.

Response：We have added the discussion of the empirical results in the empirical application section. The additional contents are as follows：

···By observing the variation of the coefficient between CG and GI it can be found that: The effect of CG on GI is as the saying "too much water drowned the miller" goes, i. e. too much CG will have a negative effect on GI. This result is consistent with the view of scholar Wang [12].CG can send positive signals to the outside world such as being financially sound or socially responsible, thus helping firms to obtain more external resource support [13]. However, if the level of CG input is low, it neither draws external attention to obtain external resource support nor takes up too much of the organization's internal resources, so it does not have a substantial impact on the firm's GI. When the level of CG input is too high, CG will overly occupy the internal resources of the organization to have a negative impact on GI. Therefore, a reasonable level of CG input is what will promote GI, and the management of CG should be emphasized at the strategic level of the organization···

···By observing the variation of the coefficient between GS and GI it can be found that: There was a positive contribution of GS to GI at different levels of CG input intensity. The result is consistent with scholar Bai's view[9] that GS promotes GI. In addition, the promotion effect of GS on GI is continuously weakened as the intensity of CG input increases, i.e., there is an optimal promotion interval of GS on GI under different CG input levels···

See L.459-470& L.483-488 of the manuscript（Revised Manuscript with Track Change）for details.

7. Authors needs to check the full text.

Response：We have rechecked the contents of the manuscript to minimize errors in presentation, spelling, etc.

Once again, thank you very much for your constructive comments and suggestions which would help us both in English and in depth to improve the quality of the paper.

Kind regards,

Hongpeng WANG

E-mail: 348557935@qq.com

---

## [Decision Letter · Decision Letter 1]

28 Nov 2022

Does charitable giving reduce firms' willingness to invest in green innovation?

PONE-D-22-18951R1

Dear Dr. Wang,

We’re pleased to inform you that your manuscript has been judged scientifically suitable for publication and will be formally accepted for publication once it meets all outstanding technical requirements.

Kind regards,

Xingwei Li, Ph.D.

Academic Editor

PLOS ONE

Additional Editor Comments (optional):

Reviewers' comments:

Reviewer's Responses to Questions

**Comments to the Author**

1. If the authors have adequately addressed your comments raised in a previous round of review and you feel that this manuscript is now acceptable for publication, you may indicate that here to bypass the “Comments to the Author” section, enter your conflict of interest statement in the “Confidential to Editor” section, and submit your "Accept" recommendation.

Reviewer #1: All comments have been addressed

Reviewer #2: All comments have been addressed

Reviewer #3: All comments have been addressed

2. Is the manuscript technically sound, and do the data support the conclusions?

Reviewer #1: Yes

Reviewer #2: Yes

Reviewer #3: Yes

3. Has the statistical analysis been performed appropriately and rigorously? 

Reviewer #1: Yes

Reviewer #2: Yes

Reviewer #3: Yes

4. Have the authors made all data underlying the findings in their manuscript fully available?

Reviewer #1: Yes

Reviewer #2: Yes

Reviewer #3: Yes

5. Is the manuscript presented in an intelligible fashion and written in standard English?

Reviewer #1: Yes

Reviewer #2: Yes

Reviewer #3: Yes

6. Review Comments to the Author

Reviewer #1: Thanks for the positive revision from authors. I agree with this manuscript and think it can be accepted now.

Reviewer #2: As for the paper's content, its structure is correct; it is easy to read; it contains all the relevant and necessary information for the reader. Therefore, I strongly recommend this article for acceptance for further publication in this reputed Journal without any more changes.

Reviewer #3: All suggestions are carefully fixed and I am very glad to recommend to publish this interesting paper.

7. PLOS authors have the option to publish the peer review history of their article (what does this mean?). If published, this will include your full peer review and any attached files.

Reviewer #1: No

Reviewer #2: No

Reviewer #3: No

---

## [Editor Report · Acceptance letter]

16 Dec 2022

PONE-D-22-18951R1 

Does charitable giving reduce firms' willingness to invest in green innovation? 

Dear Dr. Wang:

I'm pleased to inform you that your manuscript has been deemed suitable for publication in PLOS ONE. Congratulations! Your manuscript is now with our production department. 

Kind regards, 

on behalf of

Prof. Dr. Xingwei Li 

Academic Editor

PLOS ONE